# HyperVQ: MLR-based Vector Quantization in Hyperbolic Space

**Nabarun Goswami**                                                                 *nabarungoswami@mi.t.u-tokyo.ac.jp*
*The University of Tokyo*

**Yusuke Mukuta**                                                                   *mukuta@mi.t.u-tokyo.ac.jp*
*The University of Tokyo, RIKEN*

**Tatsuya Harada**                                                                  *harada@mi.t.u-tokyo.ac.jp*
*The University of Tokyo, RIKEN*

**Reviewed on OpenReview:** *https://openreview.net/forum?id=WgJgIULL9Q*

## Abstract

The success of models operating on tokenized data has heightened the need for effective tokenization methods, particularly in vision and auditory tasks where inputs are naturally continuous. A common solution is to employ Vector Quantization (VQ) within VQ Variational Autoencoders (VQVAEs), transforming inputs into discrete tokens by clustering embeddings in Euclidean space. However, Euclidean embeddings not only suffer from inefficient packing and limited separation—due to their polynomial volume growth—but are also prone to codebook collapse, where only a small subset of codebook vectors are effectively utilized. To address these limitations, we introduce HyperVQ, a novel approach that formulates VQ as a hyperbolic Multinomial Logistic Regression (MLR) problem, leveraging the exponential volume growth in hyperbolic space to mitigate collapse and improve cluster separability. Additionally, HyperVQ represents codebook vectors as geometric representatives of hyperbolic decision hyperplanes, encouraging disentangled and robust latent representations. Our experiments demonstrate that HyperVQ matches traditional VQ in generative and reconstruction tasks, while surpassing it in discriminative performance and yielding a more efficient and disentangled codebook.

## 1 Introduction

Tokenization is a fundamental component in modern data-processing pipelines, driven in part by the success of transformer-based models. While discrete data (e.g., text) can be tokenized directly, continuous data such as images and audio require specialized approaches, typically through vector quantization. Methods like VQVAE (Van Den Oord et al., 2017) generate discrete "tokens" from continuous inputs for generative modeling, and they also underpin various discriminative applications (Yang et al., 2023; Baevski et al., 2020b).

A persistent challenge in vector quantization is codebook collapse, where only a limited subset of codebook vectors is utilized. In this scenario, a few indices dominate the assignments while many remain "dead," receiving no gradient updates. This underutilization reduces the representational capacity of the model, ultimately impairing both generative and discriminative performance.

Recent research has highlighted the benefits of learning representations in non-Euclidean spaces, particularly hyperbolic spaces. Hyperbolic geometry naturally fosters compact embeddings due to its exponential volume growth with radius (Peng et al., 2021). Although hyperbolic spaces are often associated with hierarchical data, their advantages extend to general-purpose representations. Unlike Euclidean spaces, which exhibit polynomial volume growth, hyperbolic spaces promote well-separated clusters and reduce the risk of codebook

collapse. By allowing embeddings to spread efficiently within a bounded domain, the negative curvature inherent in hyperbolic spaces benefits tokenization even for non-hierarchical data such as images and audio. Compared to spherical manifolds (with positive curvature) or more complex non-Euclidean geometries (mixed curvature or SPD manifolds), hyperbolic spaces offer both intuitive exponential volume expansion and simpler analytical tools, making them particularly suitable for our proposed HyperVQ.

In this work, we exploit the properties of hyperbolic spaces to develop a robust and efficient tokenization method that addresses codebook collapse and enhances latent disentanglement. We reformulate the nearest neighbor search in vector quantization as a multinomial logistic regression in hyperbolic space, where quantized vectors serve as representative points of hyperbolic decision hyperplanes. Our approach, HyperVQ, differs from Gumbel Vector Quantization (GumbelVQ) (Baevski et al., 2020a), which uses the straight-through estimator (Bengio et al., 2013) to predict codebook vector indices. The key innovation lies in our choice of latent space and the adoption of geometrically constrained codebook vectors, eliminating the need for a separate codebook matrix. Leveraging the exponential volume growth of hyperbolic space, HyperVQ not only mitigates codebook collapse but also promotes implicit disentanglement of the latent space. Moreover, selecting a representative point on the decision hyperplane as the codebook vector enhances robustness to noise and outliers, making the training process less sensitive to codebook initialization. Our experimental results validate that the proposed method enhances discriminative performance while preserving generative quality.

The contributions of this work are summarized as follows:

- **First Realization of Hyperbolic Vector Quantization**: We introduce a vector quantization method for hyperbolic spaces—a critical building block in modern deep learning architectures.

- **Geometrically Constrained Codebook Vectors**: Our formulation yields enhanced disentanglement and compact latent representations while mitigating codebook collapse.

- **Improved Discriminative Performance**: We demonstrate that HyperVQ maintains generative performance comparable to standard VQ approaches and achieves superior discriminative capabilities.

The remainder of the paper is organized as follows: Section 2 reviews related work; Section 3 provides necessary preliminaries; Section 4 details our hyperbolic vector quantization formulation; and Section 5 presents our empirical evaluations.

## 2 Related Work

### 2.1 Discrete Representation Learning

Discrete representation of data is essential for various purposes, including compression and tokenization. While vector quantization (Gray, 1984) has been a critical component of classical signal processing methods, it has gained significant traction in the deep learning community. First proposed in (Van Den Oord et al., 2017), the VQVAE applies vector quantization to the latent space of a VAE, transitioning from a continuous to a discrete latent space. This approach offers advantages, such as learning powerful priors on the discrete space, as demonstrated by pixelCNN (Van den Oord et al., 2016).

More recently, tokenized image synthesis methods like VQGAN (Esser et al., 2021), VQGAN+CLIP (Crowson et al., 2022), and VQ-Diffusion (Gu et al., 2022) have shown impressive results in image generation. While discrete representation has excelled in generative tasks, vector quantization has also been integrated into discriminative tasks. For instance, Yang et al. (2023) proposed combining quantized representation with unquantized ones for quality-independent representation learning, and Lu et al. (2023) introduced a hierarchical vector quantized transformer for unsupervised anomaly detection. Vector quantization's applicability extends beyond computer vision to domains such as audio (Baevski et al., 2020b; Zeghidour et al., 2021; Baevski et al., 2020a).

While several works have explored enhancing vector quantization operations to address issues like codebook collapse (Łańcucki et al., 2020; Roy et al., 2018; Takida et al., 2022), to the best of our knowledge, no work has explored simultaneously improving the generative and discriminative capabilities of quantized representations.

## 2.2 Hyperbolic Deep Learning

Non-Euclidean geometry offers a promising approach to uncover inherent geometrical structures in high-dimensional data. Hyperbolic spaces, known for their exponential volume growth with respect to radius, induce low-distortion embeddings and provide better model interpretation (Peng et al., 2021). Hyperbolic embeddings exhibit robustness to noise and outliers and demonstrate better generalization capabilities with reduced overfitting, computational complexity, and data requirements (Nickel & Kiela, 2017; Khrulkov et al., 2020).

Various works, including (Ganea et al., 2018; Shimizu et al., 2021), lay the mathematical foundation for creating neural networks in hyperbolic space. Building upon these foundations, methods have been proposed to address tasks such as image segmentation (Atigh et al., 2022), audio source separation (Petermann et al., 2023), image-text representation learning (Desai et al., 2023), and variational autoencoders (Mathieu et al., 2019; Skopek et al., 2020).

While a solid foundation of hyperbolic neural network building blocks have been laid, there still lacks a principled and effective vector quantization method utilizing the hyperbolic space since it is such an important mechanism of several modern neural network architectures across domains. In this work, we explore the definition of a VQ method for the hyperbolic space which can improve feature disentanglement while maining the geometric properties of the quantized latents.

# 3 Preliminaries

In this section, we provide a concise overview of non-Euclidean geometry and relevant theory crucial for understanding the foundations of our work. While we touch upon the basics, for an in-depth exploration, we recommend referring to (Ganea et al., 2018). Additionally, we introduce the concepts of vector quantization and vector quantized variational autoencoders, laying the groundwork for the subsequent discussions.

## 3.1 Hyperbolic Geometry

### 3.1.1 Riemannian Manifold

A manifold is an $n$-dimensional topological space $\mathcal{M}$, such that $\forall x \in \mathcal{M}$, the tangent space, $\mathcal{T}_x\mathcal{M}$ of $\mathcal{M}$ at $x$, is Euclidean, $\mathbb{R}^n$. A manifold paired with a group of Riemannian metric tensors, $g_x : \mathcal{T}_x\mathcal{M} \times \mathcal{T}_x\mathcal{M} \to \mathbb{R}^n$, is called a Riemannian manifold $(\mathcal{M}, g)$.

### 3.1.2 Hyperbolic Space and Poincaré Ball Model

A Riemannian manifold is called a hyperbolic space if its sectional curvature is negative and constant everywhere. There are several models to represent the $n$-dimensional hyperbolic space with constant negative curvature, such as the Hyperboloid model, the Poincaré ball model, the Beltrami-Klein model, etc. In this work, we use the Poincaré ball model, which is defined by the pairing of the manifold $\mathbb{P}_c^n = \left\{ x \in \mathbb{R}^n \mid \|x\| < \frac{1}{\sqrt{c}} \right\}$ and the metric $g_x^{\mathbb{P}} = \lambda_x^2 g_x^{\mathbb{R}^n}$ where $\frac{1}{\sqrt{c}}$ is the radius of the ball, $g_x^{\mathbb{R}^n}$ is the Euclidean metric on $\mathbb{R}^n$ and $\lambda_x = 2(1-c\|x\|^2)^{-1}$

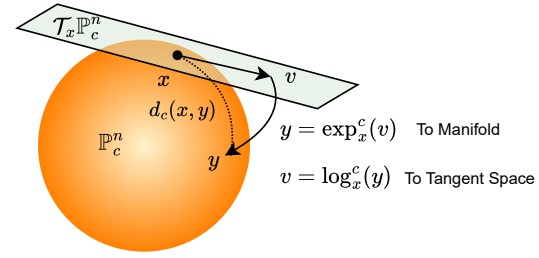

Figure 1: Illustration of the Poincaré ball model and its associated exponential and logarithmic maps

is called the conformal factor between the two metrics. Two metrics are conformal if they define the same angles.

### 3.1.3 Gyrovector Spaces and Poincaré Operations

Gyrovector spaces provide an algebraic structure to the hyperbolic space, which is analogous to the vector space structure of the Euclidean space. Several operations on the Poincaré ball model $\mathbb{P}_c^n$ are defined under this framework such as Möbius addition $\oplus_c$, distance between points on the manifold $d_c(x, y)$, exponential and logarithmic maps. We refer the reader to (Ganea et al., 2018) for a detailed explanation of these operations. Here we present the definitions of the exponential and logarithmic maps which are pertinent to our work. The following operations are defined under the framework of gyrovector spaces and considering the Poincaré ball model $\mathbb{P}_c^n$:

$$\exp_x^c(v) = x \oplus_c \left( \tanh \left( \sqrt{c} \frac{\lambda_x \|v\|}{2} \right) \frac{v}{\sqrt{c}\|v\|} \right) \tag{1}$$

$$\log_x^c(y) = \frac{2}{\sqrt{c}\lambda_x} \tanh^{-1} \left( \sqrt{c} \, \|-x \oplus_c y\| \right) \frac{-x \oplus_c y}{\|-x \oplus_c y\|} \tag{2}$$

The exponential and logarithmic maps allow us to move between the tangent space (Euclidean) and the manifold as shown in Figure 1.

### 3.1.4 Poincaré Hyperplanes, MLR and Unidirectional MLR

In (Ganea et al., 2018), the authors generalized the concept of a hyperplane in Euclidean space to hyperbolic space, by defining it as the set of all geodesics containing an arbitrary point $p \in \mathbb{P}_c^n$ and orthogonal to a tangent vector $a \in \mathcal{T}_p\mathbb{P}_c^n \setminus \{\mathbf{0}\}$.

$$H_{a,p}^c = \{x \in \mathbb{P}_c^n : \langle -p \oplus_c x, a \rangle = 0\} \tag{3}$$

However, as shown in (Shimizu et al., 2021), this definition causes over-parameterization of the hyperplane. They instead proposed a unidirectional hyperplane formulation by fixing the choice of the point on the manifold to be parallel to the normal vector. This is done by introducing a new scalar parameter $r_k \in \mathbb{R}$ which allows the definition of the bias vector in terms of $r_k$ and the normal vector $a_k$, as,

$$q_k = \exp_0^c(r_k[a_k]) \tag{4}$$

and correspondingly, the hyperplane can be defined in terms of $q_k$ as,

$$\bar{H}_{a_k, r_k}^c := \{x \in \mathbb{P}_c^n \mid \langle -q_k \oplus_c x, a_k \rangle = 0\} \tag{5}$$

Following this new formulation, the hyperbolic multinomial logistic regression (MLR), which is a generalization of the Euclidean MLR formulation (from the perspective of distances to margin hyperplanes), to perform classification tasks in hyperbolic space was defined. The unidirectional hyperbolic MLR formulation given $K$ classes, $k \in \{1, 2, \ldots, K\}$ and for all $x \in \mathbb{P}_c^n$ is defined as follows:

$$\begin{aligned} v_k &= p(y = k \mid x) \\ &\propto \operatorname{sign}\left(\langle -q_k \oplus_c x, a_k \rangle\right) \|a_k\| \, d_c\left(x, \bar{H}_{a_k, r_k}^c\right), \end{aligned} \tag{6}$$

where $d_c$ is the distance between a point and a Poincaré Hyperplane(Shimizu et al., 2021).

## 3.2 Vector Quantization and Limitations in Euclidean VQVAEs

Vector Quantization (VQ) is a widely used data compression and signal processing technique wherein a continuous input signal $z_e \in \mathbb{R}^N$ is approximated by mapping it to the nearest vector from a finite codebook $C = \{C_1, C_2, \ldots, C_K\}$. Formally,

$$k = q(z_e) = \arg\min_k \|z_e - C_k\|, \quad \hat{z}_e = z_q = C_k, \tag{7}$$

where $k$ is the index of the nearest codebook vector. VQ is central to applications such as image and speech compression, clustering, and quantization tasks, by reducing data dimensionality while preserving essential information.

Built atop this idea, the Vector Quantized Variational Autoencoder (VQVAE) introduces a discrete latent space between the encoder and decoder of a standard Variational Autoencoder. The encoder outputs a continuous latent vector $z_e$, which is quantized into $z_q$ via VQ, and subsequently passed to the decoder for input reconstruction. Although this discrete mapping is non-differentiable, end-to-end training is facilitated by a straight-through estimator that bypasses the discontinuity. The training objective typically includes a commitment loss:

$$\mathcal{L} = \log p\big(x \mid z_q\big) \; + \; \big\|\mathrm{sg}[z_e] - z_q\big\|_2^2 \; + \; \beta \,\big\|z_e - \mathrm{sg}[z_q]\big\|_2^2, \tag{8}$$

where $\mathrm{sg}[\cdot]$ is the stop-gradient operator and $\beta$ weights the commitment term. Variants of VQVAE have explored alternative codebook selection methods, such as replacing the nearest-neighbor lookup with a Gumbel Softmax sampling mechanism (Bengio et al., 2013; Baevski et al., 2020a).

Despite their widespread applicability, Euclidean VQVAEs can suffer from codebook collapse, wherein only a small subset of codebook vectors is effectively utilized. Concretely, let $\mathcal{B} = \{z_e^{(i)}\}_{i=1}^N$ be a batch of encoded samples, and define the codebook usage distribution

$$p\big(C_k\big) \;=\; \frac{1}{N}\sum_{i=1}^N \mathbb{I}\big(q(z_e^{(i)}) = k\big), \tag{9}$$

where $\mathbb{I}(\cdot)$ is the indicator function and $q(\cdot)$ is the nearest-neighbor assignment. Codebook collapse arises when $p(C_k) \approx 0$ for most $k$, meaning the majority of samples map to a small subset of codebook vectors. This tendency is exacerbated by the polynomial volume growth in Euclidean space, limiting how effectively clusters can be separated as the model's capacity or data complexity grows. Consequently, the encoder outputs can concentrate around a few dominant codebook vectors, reducing the representational capacity of the model and negatively impacting both its generative and discriminative performance.

Our work addresses these shortcomings by exploring a hyperbolic formulation of vector quantization. By exploiting the exponential volume growth of hyperbolic space, we seek to better distribute latent representations across codebook vectors and mitigate collapse, thereby enhancing the overall robustness and efficiency of the learned discrete representations.

## 4 Method

### 4.1 Overview

In an $n$-dimensional Euclidean space $\mathbb{R}^n$, the volume of an $n$-ball of radius $r$ grows polynomially as $V_{\mathrm{E}}(r) \propto r^n$, making it necessary to expand $r$ substantially to maintain clear separation among multiple clusters. By contrast, in $n$-dimensional hyperbolic space $(\mathbb{P}_c^n)$, the volume of a ball of radius $r$ satisfies

$$V_{\mathrm{H}}(r) \;\propto\; \sinh^{n-1}(r) \;\approx\; e^{(n-1)r} \quad \text{for large } r. \tag{10}$$

Hence, if we consider $k$ clusters each modeled as an $n$-ball, their total volumes scale as

$$V_{\mathrm{total,E}} \propto \; k\,r^n \quad \big(\text{Euclidean space}\big), \tag{11}$$

$$V_{\mathrm{total,H}} \propto \; k\,e^{(n-1)r} \quad \big(\text{Hyperbolic space}\big). \tag{12}$$

Because the entire hyperbolic space $\mathbb{P}_c^n$ lies within a unit ball of Euclidean radius $1/\sqrt{c}$, distances near the boundary grow exponentially, allowing more efficient packing of clusters within a bounded region while preserving separation. In Euclidean space, by contrast, one must increase the radius $r$ significantly to maintain cluster separation, resulting in less efficient packing. Additionally, the exponential volume growth also encourages balanced codebook usage. For a more detailed intuitive and theoretical discussion, refer to appendix A.

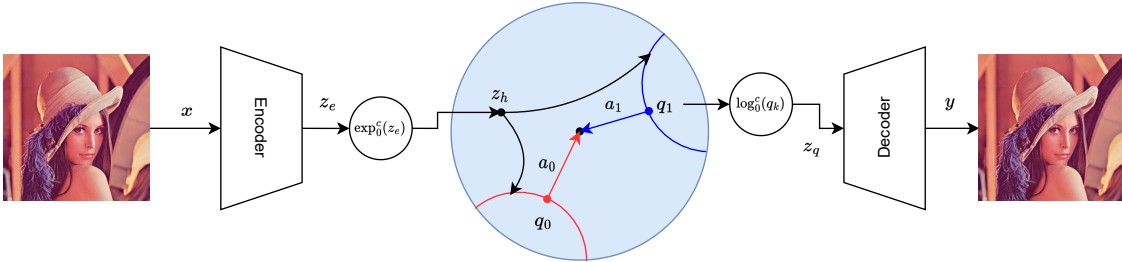

Figure 2: Illustration of HyperVQ in a VQVAE setting. The key steps are the projection of the euclidean embeddings to the hyperbolic space, classification of the projected embeddings to one of $K$ classes, and utilization of the representative point on the hyperplane, $q_k$, as the codebook vector which is projected back to the Euclidean space.

Motivated by this property, we formulate the Vector Quantization (VQ) operation as a multinomial logistic regression (MLR) directly in hyperbolic space. Specifically, rather than assigning latent vectors to the nearest codebook entry in a Euclidean sense, we perform classification in $\mathbb{P}_c^n$ using hyperbolic MLR. To further promote disentanglement, we propose using the representative points of hyperbolic decision hyperplanes as codebook vectors. Unlike an arbitrarily learned embedding matrix, these representative points capture the geometric structure more faithfully and encourage distinct separations in the latent space.

Figure 2 illustrates the overall pipeline of an encoder-decoder framework augmented with our HyperVQ. The encoder and decoder remain in Euclidean space, while the codebook lookup (i.e., quantization) occurs in $\mathbb{P}_c^n$. Algorithm 1 provides a task-agnostic training procedure, and Section 4.2 details the hyperbolic MLR formulation and the geometric constraints that define our codebook vectors. Different tasks may employ diverse encoder-decoder architectures, but the quantization mechanism in hyperbolic space stays the same, exploiting the exponential volume growth to achieve compact and efficient clustering of embeddings.

In the following subsections, we provide a detailed description of HyperVQ.

## 4.2 HyperVQ

### 4.2.1 Hyperbolic MLR for Vector Quantization

Given an encoder output $z_e \in \mathbb{R}^d$, we first obtain its hyperbolic embedding $z_h = \exp_0^c(z_e)$, where $\exp_0^c(\cdot)$ is the exponential map at the origin ( Equation (1)). We then treat the codebook selection problem as a classification ( (Baevski et al., 2020a)) in $\mathbb{P}_c^n$ via hyperbolic multinomial logistic regression (MLR). Let $\ell_k(z_h)$ be the logit for codebook index $k \in \{1, 2, \ldots, K\}$. To assign $z_h$ to one of the $K$ codebook vectors, we apply a Gumbel Softmax over these logits:

$$p_k(z_h) = \frac{\exp\big((\ell_k(z_h) + g_k)/\tau\big)}{\sum_{j=1}^{K} \exp\big((\ell_j(z_h) + g_j)/\tau\big)}, \tag{13}$$

where $\{g_1, \ldots, g_K\}$ are i.i.d. samples from the Gumbel distribution (Jang et al., 2017) and $\tau > 0$ is a temperature parameter. The codebook index is chosen by

$$k^* = \arg\max_k \, p_k(z_h). \tag{14}$$

To enable end-to-end training, we use the straight-through estimator (Bengio et al., 2013) when backpropagating through the discrete choice. This ensures gradients pass from the decoder's reconstruction loss (or downstream objectives) back to the hyperbolic logits $\ell_k(\cdot)$. We rely on the unidirectional MLR formulation (discussed in Equation (6)) for calculating these logits in a parameter-efficient manner.

### 4.2.2 Geometrically Constrained Codebook Vectors

The selection of codebook vectors is a critical factor in vector quantization. Traditional methods include choosing cluster centroids via online K-Means (as in standard VQVAE) or learning an embedding matrix (as in Gumbel Vector Quantization). While K-Means centroids reflect intrinsic geometry in Euclidean space, direct extensions to hyperbolic space require iteratively computing central measures (Lou et al., 2020), which is computationally expensive and numerically sensitive near the Poincaré ball boundary. On the other hand, embedding matrices optimized purely by a reconstruction objective may overfit to input patterns, reducing diversity among codebook entries and weakening disentanglement.

To overcome these limitations, we assign each codebook entry to be the representative point of a hyperbolic decision hyperplane, denoted by $q_k$. In particular, $q_k$ encodes both the position and orientation of the $k$-th hyperplane in $\mathbb{P}_c^n$ (see Section 3.1.4 for its unidirectional definition). Unlike a centroid-based approach that averages embeddings, $q_k$ remains stable under outliers and preserves local geometric structure. Moreover, the decision hyperplane formulation naturally enforces distinct regions in the latent space, improving disentanglement across different codebook entries.

Since the decoder typically operates in Euclidean space, we apply the logarithmic map $\log_0^c$ to project $q_k$ from $\mathbb{P}_c^n$ back to $\mathbb{R}^d$. Referring to equation 4 for $q_k = \exp_0^c(r_k[a_k])$, we have

$$z_q = \log_0^c(q_k) = \log_0^c\Big(\exp_0^c\big(r_k[a_k]\big)\Big) = r_k[a_k], \tag{15}$$

where $r_k$ is a scalar and $a_k \in \mathbb{R}^n$ represents the normal vector in the tangent space. In essence, each hyperbolic codebook vector in Euclidean space becomes the product $r_k[a_k]$. This mapping incorporates the hyperbolic geometry (via $r_k$ and $a_k$) without requiring computationally heavy centroid calculations.

An important consequence of using $q_k$ is that the decoder is forced to reconstruct from geometrically meaningful and relatively disentangled features. As a result, each codebook entry captures a distinct direction and radius in the tangent space, and the exponential growth property of $\mathbb{P}_c^n$ helps maintain clear separations between codebook vectors. This structure not only mitigates codebook collapse but also enhances discriminative performance, since the latent space is partitioned by well-defined hyperplanes rather than overlapping or redundant embedding vectors.

---

**Algorithm 1** HyperVQ Training

---

1: **Initialization:** Initial network parameters of Encoder ($\theta_E$), Decoder ($\theta_D$), and Quantizer ($\theta_Q = \{a_k, r_k\}$ for $k = 0, 1, \ldots, K$ ) and temperature, $\tau \in [\tau_{\max}, \tau_{\min}]$, and decay factor $\delta$
2: **Training:**
3: **for** each iteration, $j$ **do**
4:       sample data, $x$
5:       $z_e = \text{Encoder}(x)$
6:       $z_h = \exp_0^c(z_e)$
7:       $logits = \text{unidirectional\_mlr}(z_h)$ (Equation (6))
8:       $k = \text{argmax}(\text{gumbel\_softmax}(logits, \tau))$
9:       $z_q = r_k[a_k]$ (Equation (15))
10:      $\tau = \max\big(\tau_{\max} \cdot \delta^j, \tau_{\min}\big)$
11:      $y = \text{Decoder}(z_q)$
12:      Compute objective function: $\mathcal{L}$
13:      Update $\theta_E, \theta_D, \theta_Q$ by minimizing $\mathcal{L}$ with gradient descent
14: **end for**

---

## 5 Experiments

To assess the effectiveness of our proposed method, we conducted experiments across diverse tasks, encompassing image reconstruction, generative modeling, image classification, and feature disentanglement. The implementation details and results of each experiment are elaborated upon in the following subsections.

### 5.1 Implementation Details

Our method was implemented using the PyTorch library (Paszke et al., 2019). Model training was conducted on 4 A100 GPUs utilizing the Adam optimizer (Kingma & Ba, 2014), with a learning rate set at 3e-4 and a batch size of 128 per GPU, unless otherwise specified. For hyperbolic functions, we employed the geoopt library (Kochurov et al., 2020) and referenced the official implementation of hyperbolic neural networks++ (Shimizu et al., 2021).

### 5.2 Reconstruction and Generative Modeling

To assess the generative modeling capabilities of the proposed HyperVQ method, we conducted experiments on the Cifar100 (Krizhevsky, 2009) and ImageNet (Russakovsky et al., 2015) datasets. We trained VQVAEs with the original formulation, referred to as KmeansVQ, and our proposed hyperbolic formulation, termed as HyperVQ, for varying numbers of codebook vectors $K$. The encoder comprises an initial convolutional encoder block, followed by a stack of residual blocks, before being passed through the quantization layer. The decoder is the inverse of the encoder, with the residual blocks and the convolutional encoder block replaced by a transposed convolutional decoder block for upsampling. The reconstruction results are presented in Table 1. By comparing the reconstruction mean squared error (MSE) of the two methods, we observed that our proposed method performs slightly better or on par with the original formulation.

Table 1: Comparison of reconstruction mean squared error (MSE) for KmeansVQ and HyperVQ on Cifar100 (test) and ImageNet (validation) datasets for different codebook sizes, $K$.

| $K$ | Cifar100 (32×32) ↓ | | | ImageNet (128×128) ↓ | | |
|---|---|---|---|---|---|---|
| | 512 | 256 | 128 | 512 | 256 | 128 |
| KmeansVQ | .264±.003 | .264±.003 | **.250±.003** | **.175±.001** | **.199±.001** | .225±.001 |
| HyperVQ | **.216±.002** | **.241±.003** | .256±.003 | **.175±.001** | .202±.001 | **.217±.001** |

Following this, we trained a generative 15-layer GatedPixelCNN (Van den Oord et al., 2016) model using the quantized representations obtained from VQVAE models trained on the ImageNet dataset with $K = 512$ codebook vectors for 50 epochs.

Table 2: Generative modeling performance of GatedPixelCNN with VQVAE quantized representations on ImageNet, comparing HyperVQ and the original formulation in terms of Fréchet Inception Distance (FID) and Inception Score (IS)

| | FID ↓ | IS ↑ |
|---|---|---|
| KmeansVQ | 143.78 | 5.50 ± 0.07 |
| HyperVQ | **130.31** | **5.77 ± 0.05** |

As shown in Table 2, HyperVQ demonstrates comparative generative modeling performance compared to the original formulation, measured by the Fréchet Inception Distance (FID)(Heusel et al., 2017) and Inception Score (IS)(Salimans et al., 2016). Additionally, Figure 3a displays some reconstructions achieved with HyperVQ on the ImageNet validation set (128×128), while Figure 3b presents samples generated by the GatedPixelCNN model. We reiterate here, that he goal of this experiment is to show the effectiveness of HyperVQ and not to achieve state-of-the-art.

### 5.3 Image Classification

To substantiate our claim that HyperVQ learns a more disentangled representation, thereby enhancing discriminative performance, we conducted experiments on the Cifar100 dataset. In these experiments, we

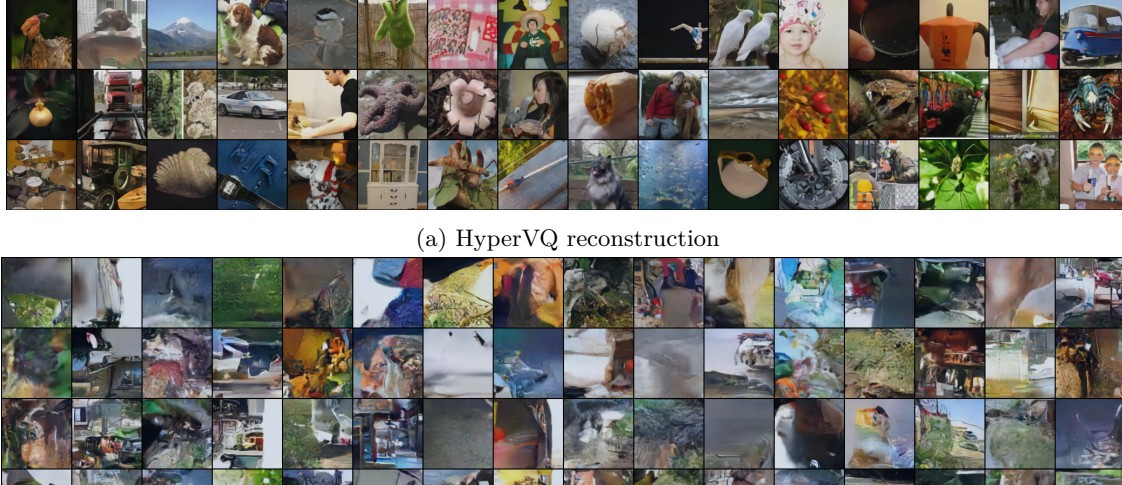

(a) HyperVQ reconstruction

(b) Samples from GatedPixelCNN trained on top of HyperVQ encodings

Figure 3: Image reconstruction and generative modeling results on ImageNet.

initially trained VQVAEs with various quantization methods and a varying number of codebook vectors, $K$. Subsequently, we employed the pre-trained VQVAE as a feature extractor and trained a lightweight classifier atop it, consisting of a convolutional layer with ReLU activation, followed by a global average pooling layer and two fully connected layers to obtain the logits. Only the classifier block is trained using the cross-entropy loss, while keeping the parameters of the convolutional encoder, residual blocks, and the quantization layer fixed. The goal of this experiment is to show the effectiveness of HyperVQ embeddings and not to achieve state-of-the-art accuracy.

For this experiment, in addition to KmeansVQ and HyperVQ, we also included GumbelVQ for comparison. All models underwent training for 500 epochs, utilizing the same settings as described in Section 5.1.

Table 3: Evaluation of discriminative performance using pre-trained encoder and quantizer of a VQVAE as feature extractors on the Cifar100 test set. Additionally, codebook usage as measured by average perplexity is reported in the last column.

| $K$ | 512 | 256 | 128 | 64 | Perplexity ($K = 512$) |
|---|---|---|---|---|---|
| KmeansVQ | 30.04±0.90 | 29.54±0.89 | 30.23±0.90 | 29.58±0.89 | 58.09 |
| GumbelVQ | 25.99±0.86 | 26.93±0.87 | 26.94±0.87 | 27.45±0.87 | 169.59 |
| HyperVQ | **31.59±0.91** | **31.06±0.91** | **30.63±0.90** | **30.14±0.90** | **220.22** |

The results presented in Table 3 indicate that HyperVQ consistently outperforms other methods in terms of classification accuracy, irrespective of the number of codebook vectors.

To gain insights into the reasons behind this improved performance, we visualize the codebook vectors learned by different methods in Figure 4 and compare the codebook usage for reconstruction based on the perplexity, defined as $\mathcal{P} = \exp\left(-\sum_{k=1}^{K} p_k \log p_k\right)$, where $p_k$ is the empirical probability of selecting the $k$-th codebook entry, on the test set as shown in Table 3. Perplexity quantifies the effective number of utilized codes, with higher values indicating more uniform usage. Additionally, we treat the codebook vector as a latent representation with $1 \times 1$ spatial dimensions and decode it using the corresponding pretrained VQVAE decoder for visualization purposes.

Insights drawn from the codebook visualization and perplexities yield several interesting observations. Firstly, it is evident that KmeansVQ exhibits low codebook usage, as illustrated in Figure 4a, where a substantial portion of the codebook vectors appears to be invalid and unused. While GumbelVQ demonstrates higher

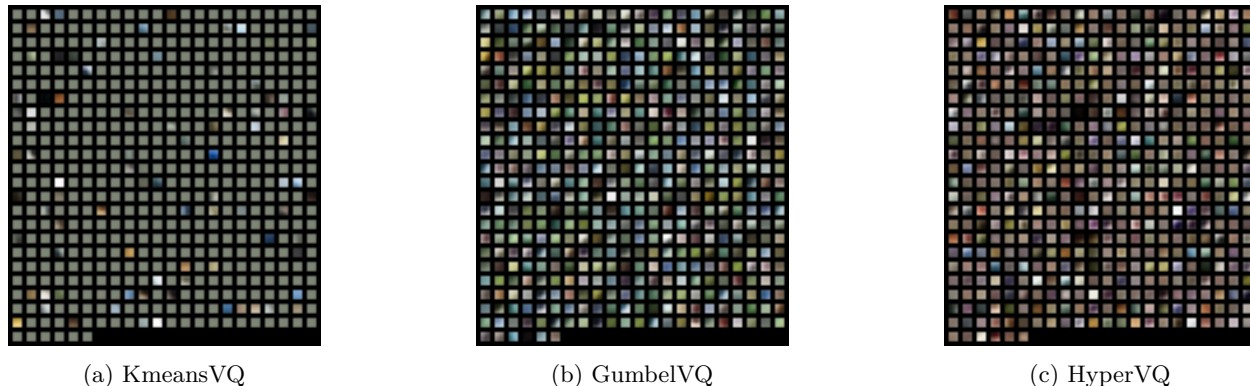

(a) KmeansVQ                    (b) GumbelVQ                    (c) HyperVQ

Figure 4: Visualization of the codebook vectors for VQVAE trained on Cifar100 with different quantization methods for $K = 512$.

codebook usage compared to KmeansVQ and learns valid codebook vectors, the vectors themselves lack strong discriminative qualities. This limitation stems from their optimization for reconstruction, leading to some redundancy in the codebook visualizations (Figure 4b), ultimately resulting in lower classification accuracy compared to KmeansVQ.

Contrastingly, the codebook vectors learned by HyperVQ exhibit both the highest perplexity (Table 3) and less redundancy (Figure 4c). This characteristic enhances the discriminative ability of the quantized embeddings, translating into the highest classification accuracy among the methods considered.

### 5.3.1 Disentanglement and Cluster Quality

To assess whether HyperVQ is capable of learning a more disentangled representation, we trained a Hyper-VQVAE model on the simple MNIST dataset. We maintained the latent dimension as 3, with 16 codebook vectors, for easy visualization without any post-processing. For comparison, we also trained a KmeansVQ-VAE model with the same configuration. To facilitate visualization, we sampled 1000 embeddings and plotted them against 3 codebook vectors each for HyperVQ and KmeansVQ.

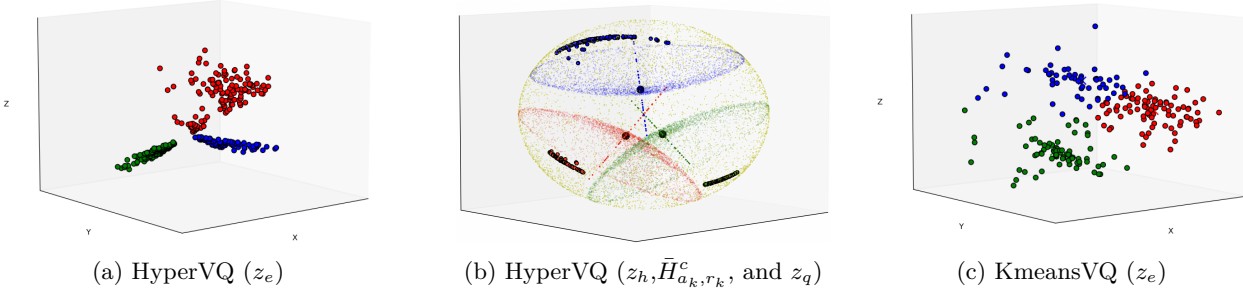

(a) HyperVQ ($z_e$)          (b) HyperVQ ($z_h$,$\bar{H}^c_{a_k,r_k}$, and $z_q$)          (c) KmeansVQ ($z_e$)

Figure 5: Vizualization and comparison of the embeddings learnt by the KmeansVQVAE and HyperVQVAE models on MNIST dataset. Figure 5a shows the pre-pojection Euclidean embeddings learnt by the HyperVQ, and Figure 5b shows the projected embeddings along with the decision hyperplanes and quantized vectors. Figure 5c shows the embeddings learnt by the KmeansVQ model,

As depicted in Figure 5, the clusters of the HyperVQVAE are more compact and disentangled than those of the VQVAE. This phenomenon arises because the hyperbolic MLR formulation encourages embeddings to be highly localized within the regions enclosed by the decision hyperplanes, thus inducing implicit disentanglement of the latent space.

Apart from the visualization, we also performed cluster quality assessment by computing the Silhoutte score (Rousseeuw, 1987) and Davies-Bouldin Index (Davies & Bouldin, 1979). To test the robustness, we

Table 4: Cluster quality and robustness assessment under standard and noisy conditions, comparing Silhouette score and Davies-Bouldin Index for KmeansVQ and HyperVQ.

|  | Standard Conditions | | Noisy Conditions | |
|---|---|---|---|---|
|  | Silhouette Score↑ | DB Index↓ | Silhouette Score↑ | DB Index↓ |
| KmeansVQ | 0.487 | 0.732 | 0.310 | 0.915 |
| HyperVQ | **0.565** | **0.553** | **0.564** | **0.560** |

applied random noise in the form of random rotations, flip, and gaussian noise and computed the above scores. From Table 4, we can see that HyperVQ is more robust towards noisy inputs and maintains its cluster compactness even under noisy conditions and in general performs better than KmeansVQ.

## 5.4 Quality Independent Representation Learning

We also applied HyperVQ to the VQ-SA model (Yang et al., 2023). This model introduces a VQ layer and a self-attention layer, combining quantized and unquantized features before the final classifier of a pretrained backbone network. This augmentation aims to enhance robustness to corruptions by learning quality-independent features. The VQ-SA model undergoes training on the ImageNet dataset, incorporating corruptions from the ImageNet-C dataset (Hendrycks & Dietterich, 2019) for data augmentation. In the original work, they employed $K = 10000$ and trained for 300 epochs. In our case, to conserve computational resources, we used $K = 1024$ and trained for 50 epochs. We also trained the original VQ-SA model under the same settings for fair comparison. Classification accuracy is reported on the clean validation set of ImageNet,

Table 5: Classification accuracy on ImageNet and ImageNet-C for the VQ-SA method with different quantization methods.

|  | Clean ↑ | Known ↑ | Unknown ↑ | mCE ↓ |
|---|---|---|---|---|
| ResNet50 | **75.704** | 37.39 | 48.94 | 76.43 |
| VQ-SA | 70.178 | 58.64 | 53.55 | 52.65 |
| HyperVQ-SA | 74.344 | **62.30** | **56.09** | **47.61** |

along with accuracy on known and unknown corruptions in the ImageNet-C dataset. Additionally, we report the mean corruption error (mCE) (Hendrycks & Dietterich, 2019). From the Table 5, we can see that the hyperVQ-SA method significantly outperforms the original VQ-SA.

## 5.5 Application to Speech Self-Supervised Learning

To explore whether HyperVQ provides performance gains beyond the domain of computer vision, we applied it to the wav2vec-2.0 (Baevski et al., 2020b) pre-training framework. In wav2vec-2.0 pre-training, intermediate convolutional features are quantized, and the transformer backbone predicts the quantization IDs from masked continuous convolutional features. The model is optimized using a contrastive loss. In our experiments, we replaced the standard vector quantization module in wav2vec-2.0 with HyperVQ and conducted pre-training on the LibriSpeech 960h dataset. As shown in Figure 6, HyperVQ achieves consistently higher codebook usage, which is reflected in the higher perplexity and resulting in a lower validation contrastive loss, thereby demonstrating the generalizability and efficacy of our proposed method. These results indicate that HyperVQ's structured representation contributes to more efficient learning dynamics, reinforcing its applicability in diverse domains beyond vision.

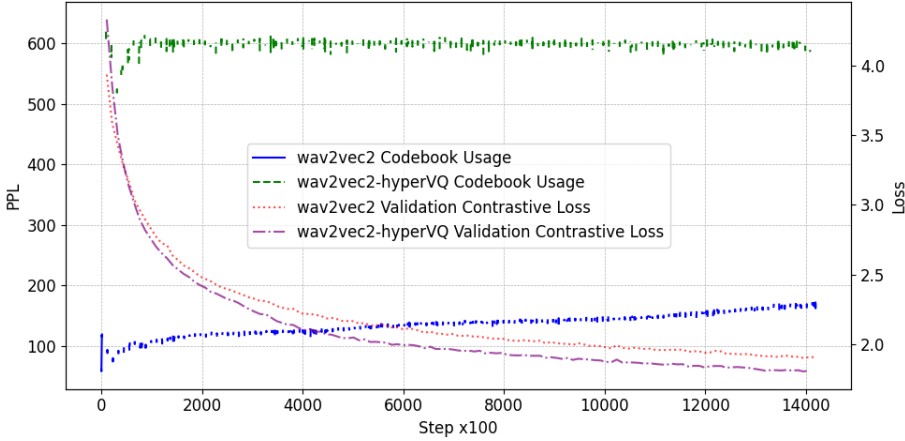

Figure 6: Application of hyperVQ to wav2vec 2.0 pre-training

## 5.6 Ablation Study

We conducted an ablation study to understand the individual and combined impacts of the hyperbolic Multinomial Logistic Regression (MLR) formulation and the use of the representative point on the decision hyperplane as the codebook vector in our proposed method.

Our evaluation compared the performance of our HyperVQ method against traditional vector quantization approaches such as KmeansVQ and GumbelVQ, as well as two variants designed for this study: HyperKmeansVQ, which uses codebook vectors in hyperbolic space learned with the KmeansVQ mechanism and distances induced by the Poincaré ball model; and HyperEmbMatVQ, which applies the hyperbolic MLR formulation but uses the embedding matrix as the codebook vectors, similar to GumbelVQ but with logits derived from the hyperbolic MLR.

Table 6: Comparison of Classification Accuracy on CIFAR-100 for various vector quantization methods with a codebook size of 512.

|  | Accuracy (%) |
| --- | --- |
| KmeansVQ | 30.04±0.90 |
| HyperKmeansVQ | 22.42±0.82 |
| GumbelVQ | 25.99±0.86 |
| HyperEmbMatVQ | 28.28±0.88 |
| HyperVQ | **31.59±0.91** |

The findings from our study, as depicted in Table Table 6, highlight the benefits of incorporating the hyperbolic MLR formulation with the embedding matrix, which notably improves classification accuracy over the GumbelVQ model. Additionally, leveraging the representative point on the decision hyperplane as the codebook vector further enhances classification performance, establishing our HyperVQ method as the superior approach among those tested.

## 6 Limitations and Conclusion

We proposed an alternative formulation for vector quantization utilizing hyperbolic space, demonstrating its effectiveness in enhancing the discriminative power of quantized embeddings while preserving the generative capabilities of the VQVAE. The hyperbolic MLR formulation was shown to encourage a more compact representation in the pre-quantized Euclidean latent space. Through various experiments, we illustrated that HyperVQ can serve as a drop-in replacement for multiple methods across domains, improving their performance.

However, it is important to note that the projections to and from the hyperbolic space exhibit sensitivity to numerical instability, particularly near the boundary of the Poincaré ball which can be circumvented by a mild radius cap (Ganea et al., 2018). However, this sensitivity still raises concerns, especially when

employing low-precision methods, such as 16-bit floating-point numbers, during model training, as it may lead to overflow and instability. Consequently, this limitation could restrict the application of HyperVQ in very large models, which are typically trained using low or mixed precision. Further analysis of the stability of HyperVQ under low-precision training is left for future work.

## Broader Impact Statement

Our work advances the methodology of vector quantization by leveraging hyperbolic space, a novel approach that can improve codebook utilization and representation disentanglement in generative models. Although primarily theoretical in its contribution, this method has practical implications for improving performance in diverse domains, including image and speech processing. However, we acknowledge that as with any advancement in generative modeling, there exists a potential risk for misuse. Improved generative models may inadvertently facilitate the creation of synthetic content that could be exploited for disinformation, deepfakes, or other unethical applications. We encourage the research community to adopt robust ethical guidelines and deploy additional safeguards when utilizing such technologies. On the positive side, our approach may enhance model efficiency and robustness, contributing to fields where reliable data representation is critical. We believe these considerations provide a balanced view of the potential societal impacts of our work.

## Acknowledgement

This work was partially supported by JST Moonshot R&D Grant Number JPMJPS2011, CREST Grant Number JPMJCR2015 and Basic Research Grant (Super AI) of Institute for AI and Beyond of the University of Tokyo.

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

# A Intuitive and Theoretical Justification for Mitigating Codebook Collapse via Hyperbolic Geometry

Hyperbolic geometry mitigates codebook collapse by leveraging its exponential volume growth. Below, we unify an intuitive explanation with a theoretical argument, showing how negative curvature promotes more uniform cluster usage than Euclidean space.

## A.1 Exponential vs. Polynomial Volume Growth

Let $\Omega \subset \mathbb{P}_c^n$ be an $n$-dimensional hyperbolic region (e.g., a Poincaré ball) with radius $R$. Up to lower-order factors, the volume of an $n$-ball in hyperbolic space, $B_H(R)$, grows as

$$\text{Vol}\big(B_H(R)\big) \approx \alpha_n\, e^{(n-1)\,R},$$

where $\alpha_n$ depends on the dimension. By contrast, in Euclidean space the volume of an $n$-ball of radius $R$, $B_E(R)$ is

$$\text{Vol}\big(B_E(R)\big) \approx R^n,$$

which grows polynomially in $R$. Consequently, for large radii, hyperbolic volumes dominate exponentially, setting the stage for more "room" at the boundaries.

## A.2 Codeword Partitions and Volume Fractions

Now let $K$ codewords partition $\Omega$ into regions $\{\mathcal{R}_1, \dots, \mathcal{R}_K\}$, defined by hyperbolic decision hyperplanes. Each codeword $k$ covers a portion of $\Omega$ with volume

$$\text{Vol}\big(\mathcal{R}_k\big),$$

so its volume fraction is

$$p_k \;=\; \frac{\text{Vol}\big(\mathcal{R}_k\big)}{\text{Vol}(\Omega)}.$$

Even a moderate extent for $\mathcal{R}_k$ implies a non-negligible $p_k$. In Euclidean space, by contrast, polynomial growth yields smaller fractions for similarly sized partitions, especially at higher dimension or larger radius.

## A.3 Chernoff Bounds and Underutilization Probability

Suppose we collect $N$ samples $\{z_h^{(1)}, \dots, z_h^{(N)}\}$ in $\Omega$, and let $X_k$ be the number assigned to codeword $k$. Under an idealized model where data are drawn i.i.d. from $\Omega$, $\mathbb{E}[X_k] = N\,p_k$. Chernoff bounds Chernoff (1952) imply that for $0 < \delta < 1$,

$$\Pr\Big[X_k \;\leq\; (1-\delta)\,N\,p_k\Big] \;\leq\; \exp\!\Big(-\tfrac{\delta^2\,N\,p_k}{2}\Big).$$

Hence, a larger $p_k$ (driven by exponential volume expansion) makes $N\,p_k$ bigger and the probability of severe underutilization exponentially smaller. This effect directly combats "codeword collapse" (dead or nearly unused codewords).

## A.4 Why Hyperbolic Partitions Tend to be More Uniform

**Boundary Sensitivity and Partition Variance.** In hyperbolic space, the volume of a region grows so fast with radius that small shifts in a cluster boundary can produce large changes in absolute volume. However, negative curvature also "pushes" these boundaries outward in a way that resists any single cluster from monopolizing the space, because other clusters can still expand at the perimeter. By contrast, in Euclidean geometry, boundary shifts can more easily let one cluster gain (or lose) a substantial fraction without the same "pushback." Consequently, the variance in partition sizes (relative to the mean size) tends to be lower in hyperbolic space, favoring more uniform distribution among clusters.

**Analogy: The "Expanding Pie".** Imagine $\Omega$ as a pie shared among $K$ people (codewords):

- Euclidean Pie (Polynomial Growth): If one person grabs a large central slice, there isn't much more pie near the edge to compensate. Others end up with smaller slices.

- Hyperbolic Pie (Exponential Growth): Even if someone starts in the center, the pie "expands" so quickly outward that there is still ample portion for others. Negative curvature effectively keeps adding new "pie" at the boundary, preventing a single cluster from hoarding too much volume.

Hence, hyperbolic geometry inherently biases the system toward more balanced clusters.

### A.5 Encoder-Generated Data and Practical Considerations

In real scenarios, latent embeddings come from an encoder (a neural network) that may deviate from i.i.d. or uniform sampling. Nevertheless, hyperbolic geometry's exponential expansion still amplifies outward spread of the data. This property inherently stabilizes codeword usage, making collapse or extreme imbalance less likely than in polynomial-growth Euclidean counterparts.

**Summary.** Combining the exponential vs. polynomial volume contrast with Chernoff bounds provides a high-level theoretical explanation for why hyperbolic geometry naturally promotes more uniform codeword utilization:

- **Exponential volume expansion** $\Rightarrow$ larger volume fractions $p_k$.

- **Higher** $p_k$ $\Rightarrow$ lower probability of underutilization by Chernoff concentration.

- **Negative curvature and boundary sensitivity** $\Rightarrow$ partitions remain closer to uniform even under moderate shifts, reducing variance in cluster sizes.

Thus, hyperbolic geometry yields an effective "self-balancing" mechanism, mitigating both codebook collapse compared to Euclidean settings.

### A.6 Numerical Experiment

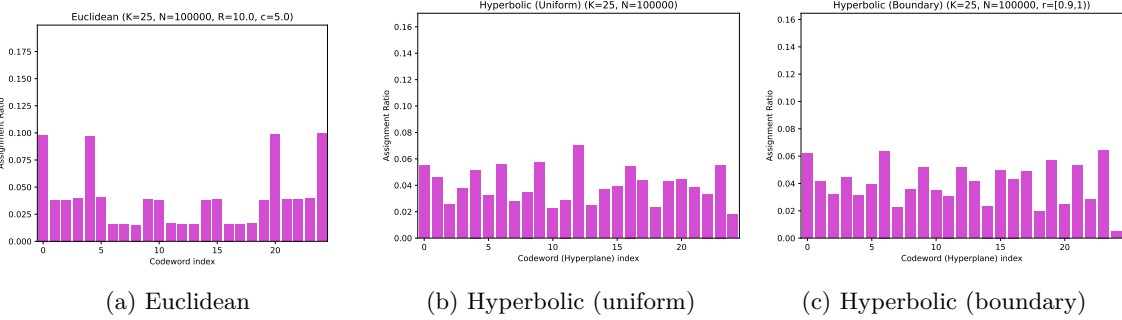

(a) Euclidean      (b) Hyperbolic (uniform)      (c) Hyperbolic (boundary)

Figure 7: Visualization of the codebook assignment ratios for toy numerical experiment.

To illustrate how hyperbolic geometry promotes more balanced codeword usage relative to Euclidean geometry, we conduct a toy experiment under controlled conditions. The key difference lies in how we place the codewords or hyperplanes before adding noise to randomize their positions.

**Setup and Codeword Initialization.**

- **Euclidean Codewords (Centers):** We define a grid in the region $[-c, c]$ and pick $K$ grid points to serve as codewords. This ensures a roughly central arrangement. To avoid perfect symmetry, we add small isotropic Gaussian noise to each coordinate. Hence, each codeword is "close to" its grid location but not placed exactly on a lattice point.

- **Hyperbolic Codewords (Unidirectional Hyperplanes):** We work in the Poincaré ball model. We place $K$ hyperplanes around the center at evenly spaced angles (e.g., $\theta_k = 2\pi k/K$), then introduce small random offsets to each hyperplane's angle, bias, and scaling parameters.

**Data Sampling and Assignment.** We sample $N$ points:

1. **Euclidean**: Uniformly in a box $[-R, R]^d$. We assign each point to the codeword that is nearest in Euclidean distance.

2. **Hyperbolic**: Uniformly in the unit Poincaré ball, and concentrating near the boundary (e.g., $\|x\| \in [r, 1)$). We assign each point to the hyperplane/codeword whose hyperbolic MLR logit is largest.

We measure the assignment ratio for each codeword $k$ (i.e. the fraction of data points assigned to it) and plot in bar charts.

**Observations** Figures 7a–7c compare the resulting codeword usage. In the Euclidean setting, a few centers often capture disproportionately large fractions of points. By contrast, the hyperbolic setting do not allow any single hyperplane to dominate—partitions stay relatively balanced. As discussed in Section A.3, the exponential volume growth boosts each codeword's volume/assignment ratio, sharply reducing the probability of codebook collapse, corroborating our theoretical justification.

