# OpenReview forum: "HyperVQ: MLR-based Vector Quantization in Hyperbolic Space"
_TMLR — Accepted by TMLR_

### Review · Reviewer_Z9Sf · 2025-02-10

**Summary Of Contributions:**

This paper introduces a novel vector quantization technique that operates in hyperbolic space. Specifically, the paper presents a hyperbolic variant vector quantization method based on hyperbolic multinomial logistic regression and the Poincaré ball model.

**Audience:**

Yes

**Claims And Evidence:**

No

**Requested Changes:**

The authors should clarify the definition of "polynomial volume growth in Euclidean space" and justify why hyperbolic space is favored over other non-Euclidean spaces for addressing codebook collapse, while also distinctly differentiating between index collapse and codebook collapse. Additionally, the paper would benefit from elaborating on the motivation for using hyperbolic embeddings to capture hierarchical data, polishing the introduction and related work sections to reduce redundancy, and addressing the numerical instability of the Poincaré ball model by either justifying its use or comparing it with alternative models. Moreover, it is essential to clarify the role of multinomial logistic regression within the proposed method and examine the implications of mapping Euclidean representations to the hyperbolic space at the origin. Finally, providing theoretical guarantees for mitigating codebook collapse, expanding the experimental evaluation to include a broader set of competing methods and updating the references to support claims regarding numerical instability will further solidify the work.

**Strengths And Weaknesses:**

# Strength
The paper is easy to read, and it’s intuitive to propose a hyperbolic variant VQ method, given the popularity of hyperbolic geometry in the recent machine learning community.

# Weakness
- The paper does not clearly explain what is meant by “polynomial volume growth in Euclidean space.” It remains ambiguous as to what this growth is measured relative to and how it integrates into the overall methodology.
- While the paper focuses on hyperbolic spaces, it does not adequately justify why hyperbolic geometry is more suitable for addressing codebook collapse compared to other non-Euclidean spaces (e.g., spherical spaces, mixed-curvature spaces, or SPD manifolds).
- The discussion on “index collapse” is vague. The paper did not clearly distinguish between index collapse and codebook collapse, and it does not explain why existing methods that address index collapse were not considered as competing approaches.
- The hyperbolic embedding induces low-distortion embedding and captures the geometric structure well only for hierarchical or tree-like structures and not for any type of data. The authors did not provide the motivation of VQ in this perspective.
- There is a noticeable redundancy between the introduction and related work sections. The introduction should focus on high-level motivations, while technical details should be confined to the related work section.
- Given the known numerical instability of the Poincaré ball model, the paper does not discuss why alternative hyperbolic models were not considered or how the performance might be impacted if another model were used.
- The importance of incorporating MLR in the proposed method is not well justified. Why not directly use the hyperbolic representation/hyperplane in VQ-VAE? or why not consider VQ in a hyperbolic VAE? why not using other classification method in hyperbolic spaces?
- The paper does not provide theoretical guarantees—such as theorems or propositions—supporting the effectiveness of the proposed method in mitigating codebook collapse.
- Mapping the Euclidean representation to the hyperbolic embedding at the origin may not necessarily capture well the hierarchical structure; how does this choice affect the proposed method?
- Although the paper references numerous works related to vector quantization in the introduction and related work sections, the experimental evaluation includes only a limited set of competing methods.
- Missing reference for numerical instability in hyperbolic spaces
- d_c in Eq. (6) is not defined

---

> ### Author Response · Authors · 2025-03-07
> **Official comment by authors (1/2)**
>
> We would like to thank the reviewer for their time and appreciate the detailed feedback and suggestions. Below, we address each of the points raised:
>
> 1. **Polynomial Volume Growth:**
>    Polynomial volume growth is defined at the beginning of Section 4.1 in our manuscript. We will add the qualifier explicitly in the sentence to ensure clarity.
>
> 2. **Justification for Hyperbolic Space compared to other non-Euclidean spaces:**
>    We have included a theoretical justification in Appendix A that explains why hyperbolic spaces with constant negative curvature are particularly effective for mitigating codebook collapse. While other non-Euclidean spaces (e.g., spherical spaces, mixed-curvature spaces, or SPD manifolds) have their merits, they lack the intuitive exponential volume growth or constant negative curvature that underpins our approach.
>
> 3. **Index Collapse vs. Codebook Collapse:**
>    We acknowledge that index collapse and codebook collapse are closely related. When only a few indices are utilized, the codebook entries of the un-utilized indices will not receive gradients and hence lead to dead codes or in other words codebook collapse occurs. Similarly, if only a handful of codes are active, then the optimization always tends to select those entries hence leading to index collapse. In our work we use the term Codebook Collapse to represent the suboptimal nature of codebook utilization.
>
> 4. **Motivation for Hyperbolic Embeddings for non hierarchical data:**
>    Although hyperbolic space is often associated with hierarchical data, its benefits extend beyond tree-like structures. In vector quantization, the exponential volume scaling in hyperbolic space ensures that quantized representations are well-separated, thereby reducing codebook collapse as demonstrated by our justification and experimental results. Our method leverages hyperbolic decision hyperplanes to enforce geometric structure in the codebook, making it effective even for non-hierarchical data (e.g., images, speech).
>
> 5. **Redundancy Between Sections:**
>    We have revised the manuscript to reduce redundancy between the introduction and related work sections. The introduction now focuses on high-level motivations, while the related work section has been covers the technical details.
>
> 6. **Numerical Instability of the Poincaré Ball Model:**
>    Given that the Poincaré and Lorentz models are isometric, switching to the Lorentz model would yield comparable performance in principle. Although the Lorentz model might offer numerical advantages in certain domains, the Poincaré model is sufficiently stable for our experiments when a mild radius cap (as proposed by HNN, Ganea et al.) is employed. Its straightforward implementation and support in existing deep learning frameworks make it a practical choice for our work. We acknowledge that exploring hybrid approaches or model switching is a promising direction for future research.

---

> > ### Author Response · Authors · 2025-03-07
> > **Official comment by authors (2/2)**
> >
> > 7. **Role of Multinomial Logistic Regression (MLR):**
> >    MLR produces a probability distribution over codebook indices, allowing us to leverage the Gumbel-Softmax reparameterization for smooth, differentiable approximations of the discrete assignments. Additionally, hyperbolic MLR measures the distance from a latent point to a decision hyperplane, which captures the geometry of the space and ensures balanced codebook usage. Our ablation study (HyperKMeansVQ in Table 6) shows that directly using the hyperbolic representation leads to significantly worse performance, underscoring the importance of our MLR formulation and the selection of the representative point.
> >
> > 8. **Theoretical Justification and Numerical Experiment:**
> >    We have added an intuitive theoretical justification in Appendix A, along with a small toy numerical experiment that verifies our claims regarding the effectiveness of our method in mitigating codebook collapse.
> >
> > 9. **Mapping Euclidean to Hyperbolic Embeddings at the origin:**
> >    Our method leverages hyperbolic space for vector quantization due to its exponential volume growth, which reduces codebook collapse and encourages compact, well-separated clusters. Unlike methods that explicitly model hierarchical structures, our approach uses hyperbolic geometry to improve the efficiency of quantization rather than enforcing a hierarchy. The choice of mapping Euclidean representations via the exponential map at the origin ensures stable and consistent transformation, supporting robust optimization. The effectiveness of this approach is demonstrated through empirical results, confirming that hyperbolic vector quantization improves representation learning without requiring hierarchical constraints.
> >
> > 10. **Comparison with Existing Methods:**
> >     Our method prevents codebook collapse through hyperbolic decision boundaries, ensuring better separation of quantized representations. Unlike existing methods that rely on heuristic techniques such as EMA, reinitialization, or regularization, our approach is optimized solely with reconstruction loss. We benchmark against methods that do not rely on additional heuristics, demonstrating that our technique effectively mitigates collapse using intrinsic geometric properties.
> >
> > 11. **Reference for Numerical Instability:**
> >     We have added a reference to HNN (Ganea et al.) to address the numerical instability of the Poincaré model and to explain the use of a simple radius cap technique in our experiments.
> >
> > 12. **Definition of $d_c$ in Equation (6):**
> >     We have clarified and defined $d_c$ in the updated manuscript to eliminate any ambiguity.
> >
> > We hope that these clarifications adequately address the reviewer's concerns and we appreciate the opportunity to improve our manuscript.

---

### Review · Reviewer_gHht · 2025-02-10

**Summary Of Contributions:**

In this paper, the authors harness the properties of hyperbolic space to develop an efficient tokenization method that improves the issue of codebook collapse and enhances disentanglement. The authors propose formulating the nearest neighbor search problem in vector quantization (VQ) as a multinomial logistic regression in hyperbolic space, with quantized vectors service as representative points of hyperbolic decision hyperplanes. They call this approach "HyperVQ" and note that the key distinctions of their method with respect to prior work lies in the choice of the latent space and the selection of geometrically constrained codebook vectors, rather than optimizing a separate codebook matrix. The paper conducts a variety of experiments to demonstrate that HyperVQ achieves generative performance on par with standard VQ approaches and also achieves improved discriminative capabilities.

**Audience:**

Yes

**Broader Impact Concerns:**

This work concerns an alternative formulation for vector quantization using hyperbolic space; given the fairly theoretical nature of this modification, I do not believe a broader impact statement specifically concerning this novelty is required. That being said, one of the main applications of the work is to generative modelling; it would be beneficial for the paper to give a brief blurb regarding the potential negative societal consequences of generative machine learning in the final section (Section 6).

**Claims And Evidence:**

Yes

**Requested Changes:**

I have highlighted my requested changes below and have specified for each whether they are critical for securing my recommendation for acceptance:

(1) I am requesting one main change in terms of results (critical for securing acceptance): please provide standard deviations for results so that the reader may assess statistical significance! This would be particularly useful in highlighting when even a small mean difference between two results is actually quite significant statistically.

(2) Although the writing in the paper is understandable and generally clear, it can and should be improved prior to publication. I am requesting the following changes to the writing in the paper (critical for securing acceptance):

- Section 2.1: The citations "Gray (1984)", "Esser et al. (2021)", "Crowson et al. (2022)" should all be parenthetical (\citep). This is a persistent issue throughout the paper, that is, in-text citations are frequently used when parenthetical citations should be used instead. Please fix this in all places where it occurs.

- Section 2.1: "Yang et al. Yang et al. (2023)" -> "Yang et al. (2023)" (duplicate text)

- Section 2.1: "Lu et al. Lu et al. (2023)" -> "Lu et al. (2023)" (duplicate text)

- Section 2.2: "Since VQ is such an important mechanism of several modern neural network architectures across domains" -> This is not a complete sentence, please fix. An easy fix may be merging this with the previous sentence.

- Section 3: "relevant theories" -> "relevant theory"

- Section 3.1.1: "a n-dimensional" -> "an n-dimensional"

- Section 3.1.2: "A Riemannian manifold is called a hyperbolic space if its sectional curvature is negative everywhere." -> This is incorrect. The sectional curvature must be both negative **and constant** everywhere.

- Section 3.1.3: Please move the word "Operations" back onto the same line with the rest of the title.

- Section 3.1.4: "an unidirectional hyperplane" -> "a unidirectional hyperplane"

- Figure 2, caption: "on the hyperplane, $q_k$ as the codebook vector" -> "on the hyperplane, $q_k$, as the codebook vector"

- Section 4.1, final paragraph: Delete the extra space before "Figure 2".

- Algorithm 1: The algorithm doesn't include the loss by which the parameters are updated via gradient descent. Either include these details or consider moving the algorithm into the appendix.

- Section 5.3: "aconsisting of a convolutional layer" -> consisting of a convolutional layer"

- Section 5.3: Please define perplexity (i.e., you are likely using this term to mean exponentiated cross-entropy here).

- Section 5.4: "outperforms the original VQ-SA" -> outperforms the original VQ-SA."

**Strengths And Weaknesses:**

## Strengths

1. The overall idea of using hyperbolic space for more efficient tokenization in the context of vector quantization is interesting and has merit; to the best of my knowledge, this is the first time hyperbolic space has been used in this way.

2. The perplexity/codebook utilization aspects of the model seem to be quite promising; in particular, I like the visualizations of the codebook vectors in Figure 4 (where it clearly seems the model improves over at the very least KmeansVQ).

## Weaknesses

1. Results-wise, I do not believe a very convincing case has been made for the utility of the model, i.e. there does not seem to be a particular "essential" application whose needs are served by this model design and by many downstream metrics HyperVQ does not seem to give a considerable improvement over KmeansVQ (e.g. see Table 1 for 256 and 128 codebook vectors) by mean difference. Perhaps most crucially, for almost all results, standard deviations are not given! This make it very hard to assess statistical significance of the results in e.g. Table 3.

2. Although this idea is interesting at a high-level, I think it is currently under-explored. I would have liked to see a more thorough investigation and some variation on the implementation of the VQ approach that perhaps goes a little outside of the "VAE" setting the paper builds on directly. For example, instead of using hyperbolic MLR for quantization and the VQVAE loss, how would a more straightforward quantization approach work, where we perform quantization by selecting the closes codebook vector by hyperbolic distance? This seems like it would make for a simple and interesting baseline that may help better justify some of the design decisions currently being made.

## Verdict

This paper presents a vector quantization approach using hyperbolic space in the VQVAE framework. To the best of my knowledge, this is a reasonably novel application of hyperbolic space and seeks to improve the issues of codebook collapse and enhances disentanglement. I believe the claims made in the paper are somewhat well-supported by the empirical results as of right now, but will be better supported once standard deviations are provided for most numerical results (see requested changes below). I also believe this paper is of interest to at least some portion of TMLR's audience, specifically those who work on either vector quantization or hyperbolic machine learning. As a consequence, I recommend an accept rating for this paper pending the completion of the requested changes below.

---

> ### Author Response · Authors · 2025-03-07
> **Official comment by authors (1/1)**
>
> We thank the reviewer for the insightful comments and suggestions. Below, we detail our responses to the points raised:
>
> **Experimental Results and Statistical Significance**
>
> 1. **Confidence Intervals:**
>    We have added 95% confidence intervals to all relevant numerical results. This update provides a clearer indication of the statistical significance of our findings, addressing concerns about the robustness of the reported improvements.
>
> 2. **Utility and Codebook Usage:**
>    While our experiments show that HyperVQ provides improvements as a drop-in replacement for standard VQ, specifically our results demonstrate a significant increase in codebook usage and a reduction in codebook underutilization. We have further supported these claims with an in-depth intuitive and theoretical justification included in Appendix A1 (per reviewer Z9Sf’s suggestion). Moreover, experiments across different modalities (images and speech) highlight the generalizability of HyperVQ and open avenues for future exploration in hyperbolic vector quantization.
>
> 3. **Ablation Study – HyperKMeansVQ Baseline:**
>    The ablation study includes HyperKMeansVQ, which implements a straightforward quantization approach by selecting the closest codebook vector by hyperbolic distance. As shown in Table 6, this naive implementation does not perform as well as HyperVQ, thereby justifying our design choices.
>
> **Writing and Formatting Corrections**
>
> We have revised the manuscript to address all noted writing issues:
> - Corrected in-text citations to parenthetical style (using \citep).
> - Removed duplicate text in the citations.
> - Revised incomplete sentences and grammatical issues.
>
> All these modifications have been incorporated throughout the manuscript.
> We have also included a short broader impact statement to the updated manuscript.
>
> ---
>
> We appreciate the reviewer's feedback, which has significantly contributed to improving both the technical and presentational quality of our paper. Please let us know if further clarifications are needed.

---

### Review · Reviewer_6v75 · 2025-02-23

**Summary Of Contributions:**

This paper presents a new method for quantizing continuous data using hyperbolic gemetry. In particular, the paper suggests, training an encoder, then doing classification in hyperbolic space to learn the codebook representation. The autoencoder is trained using reconstruction error.

The idea of using hyperbolic space is interesting, the representations learned to appear to be more diverse and contain more information.

Overall, I am for eventual acceptance of the paper, however, the current version has a variety of mathematical issues and needs a lot more experimental details.

**Audience:**

Yes

**Claims And Evidence:**

No

**Requested Changes:**

I would like the authors to answer the questions in the weakness section. In general, I would like the paper to contain more experimental details and be more precise regarding the mathematical definitions and the model.

**Strengths And Weaknesses:**

**Strengths**

This paper continues the trend of adapting Euclidean models to hyperbolic geometry. The proposed method seems to learn good representations.

**Weakness**

1. Mathematical concern.

Section 3.1.1. - Local neighborhoods can be approximated by Euclidean space. The tangent space is Euclidean.

Section 3.1.2. - $c$ is not the radius. The radius is $\frac{1}{\sqrt{c}}$ based on the equation. Hence there is a mismatch. The $c$ should also feature in the Reimannian metric. Specifically, $\lambda_x = \frac{2c}{1 - c \|x\|^2}$. This differ from Ganea et al 2018 by a factor of c. Based off of this the exp and log need to checked.

Section 3.1.4 - What is $z_h$ in Eqaution (6)? How are the logits $\ell_j$ calculated? It says that equation (6) is used, but that equation is not clear. For the logits, should the $x$ also be a $z_h$?

What are "we propose using the representative points of hyperbolic decision hyperplanes" (bottom of page 5). Again in section 4.2.2. this is referenced as the $q_k$ from the hyperbolic regression problem. However, how are there $q_k$ choosen?

2. Experimental Questions.

Does Table 1 represent training data reconstrution MSE or test data reconstruction MSE? What are the errorbars or the uncertaininity in these numbers?

The generative model, (and the decoder) are Euclidean models, do we just treat the hyperbolic embeddings as Euclidean vectors?

---

> ### Author Response · Authors · 2025-03-07
> **Official comment by authors (1/1)**
>
> Thank you for your valuable feedback. We have revised the manuscript to address both the mathematical and experimental issues raised. Below, we summarize our changes and clarifications:
>
> **Mathematical Concerns**
>
> 1. **Section 3.1.1 – 3.1.2**
>    We acknowledge the mismatch and have updated the manuscript.
>
> 2. **Section 3.1.4 – Equation (6) and Logits Calculation**
>    Equation (6) has been corrected so that both instances are consistently denoted as $x$. Subsequently, $x$ is set to $z_h$ when computing the logits $l_j$. In our implementation, the parameters of $q_k$ are initialized following the official implementation of HNN++ (Shimizu et al.): $a_k$ is drawn from a normal distribution and $r_k$ is initialized to zero. These parameters are updated during training.
>
> **Experimental Concerns**
>
> 1. **Table 1 – Reconstruction MSE and Confidence Intervals**
>    We have updated the manuscript to clarify that Table 1 reports reconstruction MSE on the test data, specifically using the CIFAR100 test set and the ImageNet validation set. Moreover, 95\% confidence intervals have been added to the reported metrics to provide a clear indication of the uncertainty in the results.
>
> 2. **Generative Model and Decoder Input**
>    As detailed in Equation (15), the input to the decoder is the Euclidean code vector computed as the $\log_0^c(q_k)$. Since $q_k$ is obtained by applying the $\exp_0^c$ map to $r_k[a_k]$, this simplifies directly to $r_k[a_k]$, as these parameters reside in Euclidean space.
>
> ---
>
> We believe these revisions adequately address the raised issues and enhance the clarity and precision of our manuscript. Thank you again for your constructive feedback.

---

### Author Response · Authors · 2025-03-07
**Updated Manuscript**

Dear Reviewers,

We sincerely appreciate your invaluable comments, which have helped us improve and refine the manuscript.

We have incorporated the requested changes, and they are marked in blue for easy visibility.

---

### Decision · Action_Editor_rRbX · 2025-04-08

**Recommendation:** Accept as is

**Comment:**

We concur with the reviewers' unanimous support for acceptance. The paper is expected to be of broad interest to the TMLR audience, and the central claim of the viability of a hyperbolic-based quantisation strategy has been well demonstrated on a range of datasets. The authors did a good job of addressing several questions and suggestions from reviewers, which have strengthened the paper.

**Audience:**

Reviewers were unanimously confident that the paper would be of interest to a subset of the TMLR community. Quantisation methods are of broad interest, and new techniques to resolve fundamental problems (e.g., codebook collapse) are expected to be of interest to the community. The present paper introduces a new perspective based on hyperbolic spaces, which could also inspire follow-on research efforts.

**Claims And Evidence:**

The paper's main claim is that vector quantisation may be effectively performed in hyperbolic rather than Euclidean space, leading to benefits over classical strategies (e.g., mitigating codebook collapse). This is achieved by converting a Euclidean encoder's output into hyperbolic space, and casting the problem of codebook selection as learning a multinomial logistic classifier in hyperbolic space.

Reviewers were unanimous in finding the paper's claims to be well-supported by the algorithmic and empirical evidence (following some clarifications in the author response). One reviewer was unsure if the experiments demonstrated a _significant_ edge of HyperVQ over existing quantisers; however, the reviewer was satisfied on the demonstration of the core claim of the viability of this strategy.